# Strategies to Overcome Resistance to Immune-Based Therapies in Osteosarcoma

**DOI:** 10.3390/ijms24010799

**Published:** 2023-01-02

**Authors:** Claudia Maria Hattinger, Iris Chiara Salaroglio, Leonardo Fantoni, Martina Godel, Chiara Casotti, Joanna Kopecka, Katia Scotlandi, Toni Ibrahim, Chiara Riganti, Massimo Serra

**Affiliations:** 1Osteoncology, Bone and Soft Tissue Sarcomas and Innovative Therapies, IRCCS Istituto Ortopedico Rizzoli, 40136 Bologna, Italy; 2Department of Oncology, University of Torino, Via Santena 5/bis, 10126 Torino, Italy; 3Department of Experimental, Diagnostic and Specialty Medicine (DIMES), University of Bologna, 40126 Bologna, Italy; 4Laboratory of Experimental Oncology, IRCCS Istituto Ortopedico Rizzoli, 40136 Bologna, Italy

**Keywords:** osteosarcoma, prognosis, immunotherapeutic target, tumor immune microenvironment

## Abstract

Improving the prognosis and cure rate of HGOSs (high-grade osteosarcomas) is an absolute need. Immune-based treatment approaches have been increasingly taken into consideration, in particular for metastatic, relapsed and refractory HGOS patients, to ameliorate the clinical results currently achieved. This review is intended to give an overview on the immunotherapeutic treatments targeting, counteracting or exploiting the different immune cell compartments that are present in the HGOS tumor microenvironment. The principle at the basis of these strategies and the possible mechanisms that HGOS cells may use to escape these treatments are presented and discussed. Finally, a list of the currently ongoing immune-based trials in HGOS is provided, together with the results that have been obtained in recently completed clinical studies. The different strategies that are presently under investigation, which are generally aimed at abrogating the immune evasion of HGOS cells, will hopefully help to indicate new treatment protocols, leading to an improvement in the prognosis of patients with this tumor.

## 1. Introduction

Improving the prognosis and cure rate of HGOSs (high-grade osteosarcomas) is still challenging, despite several innovative therapeutic approaches having been explored in the last two decades, especially to treat recurrent and relapsing patients, who experience a very poor and unsatisfactory outcome [1].

On this background, a number of preclinical and clinical studies have focused on the TIME (tumor immune microenvironment), with the aim of gaining information on candidate therapeutic markers, which may drive the planning of innovative therapeutic strategies [2,3]. TIME is a complex system that not only includes tumor cells but also many other elements, among which tumor-infiltrating immune cells and stromal cells have been reported to play relevant roles in tumor evolution [4,5]. Immune cells basically consist of innate immune cells (TAMs, tumor-associated macrophages, and MDSCs, myeloid-derived suppressor cells) and adaptive immune cells (T and B lymphocytes) [6]. TAMs and T lymphocytes are among the most abundant populations infiltrating HGOS-associated TIME [7,8]. Therefore, immunotherapeutic treatment strategies targeting, interfering or employing the different immune cell compartments, which are present in HGOS TIME, have been increasingly taken into consideration to develop new therapies, in particular for metastatic, relapsed and refractory patients.

In this review, we provide an overview of the different cell populations infiltrating the HGOS TIME, detailing their role in sustaining tumor growth and progression.

We have critically reviewed the most recent immune signatures of possible prognostic value and/or predictive of response to current treatments, which are derived from the characterization of HGOS TIME. Moreover, we have focused on the main mechanisms that induce an immunosuppressive TIME in HGOSs by limiting the activation of antitumor immune cells and by inducing an ICD (immunogenic cell death), in order to explain the low success rate of different immunotherapy treatments based on ICIs (immune checkpoint inhibitors) or CAR (chimeric antigen receptor) T cells.

Starting from these observations, we have discussed how these resistant traits can be transformed into Achilles’ heels, which may improve both immune killing by the host immune system and efficacy of adoptive immunotherapy. The review of the latest immunotherapy-based clinical trials will provide the reader with detailed insights into new therapeutic options for HGOS.

## 2. The Osteosarcoma Immune Environment

### 2.1. The Variegate Scenario of Osteosarcoma Immune Environment

The immune environment of HGOS is extremely varied and includes several cell types with either antitumor or protumoral properties (Figure 1 and Figure 2). For these reasons, specific immune signatures have been correlated with patient prognosis and specific immune populations have been exploited as therapeutic tools [9].

About 50% of immune cells present in HGOS TIME are TAMs, with a decreased ratio between antitumor M1/protumor M2 TAMs [2] (Figure 1A). The higher polarization toward the tumor-permissive M2 population relies on multiple mechanisms that often involve a crosstalk between tumor cells and TAMs. For instance, the activation of EGFR (epithelial growth factor) promotes tumor growth and TAM expansion, which further boost HGOS progression [10], leading to the hypothesis that EGFR signaling in HGOS cells releases secreted factors that favor the expansion of tumor-permissive M2 TAMs. HGOS cells overexpress the noncoding RNA RP11-361F15.2, which activates CPEB4 (cytoplasmic polyadenylation element binding protein 4), a protein with pleiotropic functions, including the induction of M2 polarization [11]. The downregulation of Notch signaling in both HGOSs and TAMs induces M2 polarization and tumor expansion [12], and the reduced production of cytokines from T-helper 1 (Th1) CD4^+^ T lymphocytes had the same effects [9]. Accordingly, the increased production of Th2-derived cytokines as interleukins IL-4 and IL-13 favors the polarization toward M2 TAMs [13]. The redundancy of multiple mechanisms underlying the expansion of M2 TAMs makes therapeutic interventions very difficult, because blocking all these pathways is virtually impossible and implies severe off-target effects. M2 TAMs promote tumorigenesis [14], stemness maintenance [15], HGOS metastatization [16] and immunosuppression on the activity of tumor-infiltrating lymphocytes (TILs) by releasing immunosuppressive soluble factors as IL-10, TGF-β2 (transforming growth factor-β2) and CCL22 (chemokine (C-C motif) ligand 22) [2]. Notably, efferocytosis, i.e., the TAM-induced phagocytosis of apoptotic cells that may durably eradicate tumor cells, is impaired in HGOSs by the expression of the tyrosine kinase MerTK receptor on both tumor cells and TAMs. In the latter, MerTK activates the p38/STAT3 pathway, which favors M2 polarization and the overexpression of the ICP (immune checkpoint) PD-L1 (programmed death-1 ligand) [17]. This event can further boost tumor progression by engaging PD-1 (programmed death-1) present on CD8^+^ TILs and inducing a strong anergy in this population.

If M2-TAMs are protumorigenic, the proinflammatory M1 macrophages reduce tumor growth [2] and may have a great immune-killing potential against HGOS. A recent study demonstrated that the coculture of M1 TAMs and HGOS cells has deleterious effects on the tumor cells via paracrine mechanism: HGOS cells secrete L-galectin 3 soluble binding protein 3 (LGALBP), which binds its receptor on M1 TAMs and activates the Akt/HSF1 (heat shock protein transcription factor 1) pathway, leading to the release of HSPA1L (heat shock protein family A member 1 like). The latter induces apoptosis of HGOS cells via interleukin 1 Receptor-Associated Kinase (IRAK) IRAK1/IRAK4 signaling [18], suggesting that the mechanism is active in inflammatory settings, i.e., when M1 TAMs predominate and secrete high amounts of IL-1, stimulating IRAK signaling. This elegant model, which unveils several druggable circuitries, however, has been realized in ex vivo cocultures. The strong therapeutic opportunities unveiled by this circuitry need further confirmation in vivo, in syngenic or humanized mice models of HGOS.

Notwithstanding TAMs being the most abundant population in HGOS TIME, no robust correlations between M1/M2 ratio and patients’ prognosis have been reported. Only the total number of TAMs—but not their polarization ratio—seems to have a favorable prognostic value in adult patients [2]. By contrast, in hypoxic pediatric HGOSs, a high number of TAMs, irrespective of their polarization, is associated with a worse outcome [19]. These contradictory findings can be biased by the small number of patients analyzed in each study, as well as by the different grade and pathological classification. Based on the results available, the role that M2 TAMs and TAMs generally play in HGOS progression should be resized.

Similarly to TAMs, TANs (tumor-associated neutrophils) have been reported within HGOS TIME, with a similar polarization into TAN1 and TAN2 [9]. Although the characterization of TAN polarization has not been investigated, they have been associated mainly with antitumor functions, since they are lower in metastatic HGOSs than in nonmetastatic ones [20].

A third myeloid-derived population, highly abundant in HGOS TIME, is represented by DCs (dendritic cells) (Figure 1B). In early stages, DCs promote phagocytosis/efferocytosis, followed by activation of TILs. By contrast, in late-stage HGOS development, phagocytosis-resistant cells emerge [21], and DCs paradoxically turn themselves into protumorigenic cells with several mechanisms. First, DCs become responsive to glutamate released by HGOS cells, and through the glutamate metabotropic receptor 4, they release the protumorigenic IL-23 and decrease the antitumor IL-12 [22]. Second, DCs themselves autocrinally produce CCR7 (C-C chemokine receptor type 7), which favors the DC recruitment but also the metastatic properties of HGOS cells [23], acting as a double-edged sword. Third, DCs are rich in PD-L1 [24]; therefore, they can contribute to induce an anergic phenotype in T lymphocytes. Overall, the positive or negative prognostic functions of DCs are still a matter of debate, and contrasting results may depend on the high inter- and intra-patient variability of DCs and on their dynamic changes during HGOS development.

The second crucial populations infiltrating TIME and playing a key role in HGOS progression are CD4^+^ and CD8^+^ TILs (Figure 2A). It is a common observation that both CD4^+^ and CD8^+^ are more abundant in metastatic or recurrent HGOSs [25], but most of these TILs are anergic because of the high expression of ICPs such as PD-1, LAG-3 (lymphocyte activating 3) and TIM-3 (T-cell immunoglobulin domain and mucin domain 3) and immunosuppressive enzymes such as IDO (indoleamine 2,3 dioxygenase), producing kynurenine [26] and subtracting tryptophan, which is an essential aminoacid for lymphocyte proliferation. The prevalence of M2 TAMs also induces TIL anergy, as demonstrated by the reactivation of immune killing mediated by TILs after the removal of CD163^+^ TAMs [26].

A classification of lymphocyte subsets in HGOSs indicated that the presence of CD4^+^ and CD8^+^ TILs, particularly of CTLs (cytotoxic T lymphocytes, e.g., CD8^+^TIA^+^ T cells), were associated with a significantly better outcome in HGOSs, while B lymphocytes, although present, apparently do not have any prognostic role [27]. The correlation was particularly strong in HGOSs expressing CD44, a homing molecule for hematological cells; conversely, there was no correlation between the infiltration of CTLs and other negative prognostic markers of HGOSs, such as ABCB1/P-gp (ATP Binding Cassette Subfamily B Member 1/P-glycoprotein) or mutated p53. Notwithstanding the lack of correlation with known markers of aggressiveness, this study is important because it provides the first indication that the HGOS molecular phenotype, in this case CD44 expression, may influence the recruitment of the immune cells and regulate immunosurveillance.

Several other molecular phenotypes of HGOS have an opposite effect on CTL recruitment. For instance, HGOSs with high levels of c-Myc are poorly infiltrated with CTLs, and most of them are anergic. Epigenetic inhibition of c-Myc expression using the bromodomain inhibitor JQ-1 restores the recruitment of CTLs and their interactions with DCs through the increase in CD40 and CD40L in CTLs and DCs, respectively, as well as the production of specific CTL clones against HGOS cells after their interaction with DCs [28]. This work sheds light on another molecular determinant—c-Myc—reducing the activation of CTLs within HGOSs, but at the same time provides an innovative epigenetic strategy to rescue CTL tumoricidal activity.

SGLT2 (sodium-glucose transporter 2), expressed at high levels in HGOSs, also contributes to immunosuppression because it downregulates the transcription of STING (stimulator of interferon genes) by activating Akt-dependent pathways. Consistently, the inhibition of SGLT2 restores the STING/IRF3/IFN-β (interferon regulatory factor 3/interferon-β) axis, which increases the recruitment of CD4^+^ and CD8^+^ TILs [29]. This work provides the first evidence that targeting HGOS metabolism, namely by interfering with glucose uptake, may be beneficial in overcoming HGOS-induced immunosuppression. On the other hand, not all the molecular features characterizing HGOSs negatively affect TILs. For instance, the activation of TLR4 (Toll-like receptor 4) on HGOS cells dramatically increases CD8^+^ TILs and reduces both tumor growth and lung metastasis in immunocompetent mice, improving the HGOS outcome [30].

Differently from T lymphocytes, B lymphocytes are present within the HGOS immune environment, although apparently they do not have any prognostic role [27], and their role in HGOS progression is still an open question [9].

NK (natural killer) cells, a powerful tumoricidal cell population, were also found within HGOSs, although some studies reported their anergy, caused by the TGF-β abundantly secreted by tumor cells, M2 TAMs and MDSCs. TGF-β inhibits the activation of NKG2D (Natural Killer Cell Lectin Like Receptor K1) receptor [31], a key step in the tumoricidal program of NK cells. Moreover, NK cells express PD-1 and are anergic, as demonstrated by the fact that agents targeting the PD-1/PD-L1 axis restore the cytolytic abilities of NK cells, e.g., the release of granzyme B [32].

Differently from other T lymphocytes, γδ T cells are a subset of lymphocytes that elicit immune-killing in a MHC-independent way. They share with NK the expression of NKG2D activating receptor and the same mechanism of tumor killing [33]. Although several studies have been performed on the efficacy of γδ T cells against bone marrow tumors such as multiple myeloma [34], only a few studies have focused on HGOSs. Only one study, however, shows that the DNA-demethylating agent, decitabine, increases the expression of NKG2D and the killing of HGOS cells by γδ T cells [33], opening a new approach to exploit this lymphocyte subset as possible adoptive immunotherapy against HGOSs.

As anticipated, the immune surveillance is determined by a balance between antitumor and tumor-permissive cells. Besides M2 TAMs, MDSC and Treg (T-regulatory) cells are the most abundant in the HGOS niche [9].

MDSCs play a physiological role in bone resorptions by activating osteoclasts via HIF-1α (hypoxia inducible factors-1α)-dependent pathways [35]. The latter promotes the release of soluble factors such as TGF-β, VEGF (vascular endothelial growth factor) and its analogue BV8, bFGF (basic fibroblast growth factor), MMP-9 (matrix metalloproteinase-9) and osteopontin, which favors tumor cell invasion within the bone, as it occurs for the metastases of several solid tumors, or the growth of HGOSs. A two-way circuitry is built in the HGOS microenvironment, because tumor cells secrete trophic factors for MDSCs as IL-1, IL-6, IL-10, TNF-α (tumor necrosis factor-α) and MCP-1 (monocyte chemoattractant protein 1) [35], receiving as a cashback an immune niche favorable to their growth.

As far as Treg cells are concerned, they have been detected in HGOS TIME, but whether they have a negative prognostic value has not been studied yet. Interestingly, one study in a murine model of HGOS reported that the administration of anti-PD-1 decreased Treg intratumor infiltration, increasing by contrast the amount of CD8^+^ CTLs. Both mechanisms likely concur to HGOS growth reduction [36], indirectly suggesting that Treg cells control CTL anergy through the PD-1/PD-L1 axis, opening a new field to investigate this unknown role of Treg cells in HGOS and new treatment opportunities.

The crosstalk between the different immune populations is sustained by cell–cell contacts and/or by soluble factors, such as cytokines and chemokines (Figure 2B). Besides the cytokines previously mentioned [9], there is a variegate panorama of less-studied cytokines and chemokines with peculiar roles in the HGOS immune environment.

For instance, IL-34 has been proven to promote HGOS growth and M2 TAM recruitment [37]. CCL18 (Chemokine C-C motif ligand 18) is secreted by M2 TAMs and promotes tumorigenesis [38]. Notably, not only the cytokines present within HGOS TIME but also the circulating ones can contribute to immunosuppression. Systemic IL-35 suppresses the activity of circulating CTLs in HGOS patients [39], producing a population that has a lower chance to exert tumoricidal effects. Conversely, among the antitumorigenic cytokines, IL-12 is associated with longer HGOS survival and increased accumulation of CD8^+^ TILs, thanks to its ability to reduce immunosuppressive populations, such as Treg cells, MDSCs and M2 TAMs, by modulating the levels of CXCL9 (C-X-C Motif Chemokine Ligand 9), CXCL2 and CCL22, which expand immunosuppressive populations [40]. Interestingly, high levels of the complement C1q factor are also associated with a better prognosis in HGOS patients [41]. Although the mechanisms are not clear, C1q is associated with increased numbers of infiltrating TAMs, CD4^+^ and CD8^+^ T lymphocytes, and memory B cells, and with decreased expression of antimetastatic genes [41]. On the other hand, the activation of the complement cascade, caused by the low/undetectable levels of restraining factors CD46, CD55, CD59 and properdin in U-2OS cells, is associated with more pronounced angiogenesis and HGOS cell growth [42].

Finally, it is noteworthy to remember that cytokines or immunomodulatory factors can be transported by exosomes released by HGOS cells. For instance, they carry TGF-β, α-fetoprotein and HSPs (Heat shock proteins), impairing TILs proliferation and activation, promoting instead the expansion of Treg cells [43] and favoring M2 polarization [44]. Exosomes can also carry ICPs or induce their expression in the recipient T cells, enlarging the immunosuppressive effects; by transporting TIM-3 to DCs and NK cells, they impair the tumoricidal activity of these populations [45]. Interestingly, in response to doxorubicin, a massive release of exosomes carrying IL-1β is observed; the transfer of these exosomes to recipient cells determines the upregulation of PD-L1 [46], creating the premise for the CTLs’ anergy.

Although it is difficult to investigate, clarifying the specific soluble factors involved in the crosstalk between different immune populations may help us to understand how the prevalence of specific populations in TIME may support HGOS progression. At the same time, such in-depth dissection of paracrine circuitries of HGOS TIME may pave the way to an array of novel tools, as neutralizing antibodies or soluble competitive analogues, as a future treatment to overcome immunosuppression typical of HGOS TIME.

### 2.2. Can the Immune Environment Characterization Be Used to Build Prognostic or Predictive Signatures?

In the last five years, with the increasing number of public accessible databases, the advances in deep sequencing [47] and the availability of the first mathematical models to predict the evolution of HGOSs [21], several studies have attempted to provide a clinical meaning to specific immune signatures detected in HGOS patients. Below, we discuss the most recent and significant evidence derived from big data analysis of mathematical algorithms designed to predict the HGOS’s evolution or response to treatment, based on the immune signatures detected in patients’ cohorts.

A ssGSEA (single-cell gene enrichment set analysis) coupled with the WGCNA (weighted gene coexpression network analysis) of 28 patients revealed that metastatic patients have a significantly low number of TAMs and Th2 lymphocytes [13]. In this cohort, MSR1 (macrophage scavenger receptor 1) and TLR7, promoters of M2 TAM polarization [48,49], emerged as the key genes determining a metastatic evolution in HGOS patients [13], pointing out the importance of M2 TAMs as associated with advanced and aggressive stages. Similarly, the immune-related DEGs (differentially expressed genes) were analyzed from UCS Xena and GEO databases, to produce an immune-score with a particular attention to TAM-related genes. A three-gene signature—TYROBP/TLR4/ITGAM (TYRO protein tyrosine kinase binding protein/Toll-like receptor 4/integrin subunit α M)—was identified as predictive of the TAMs’ polarization and used to predict an immune risk score, based on the ESTIMATE and xCell algorithms, enriched with the analysis of Hallmark and Kegg pathways. This classification allowed for the discrimination of patients at low risk, with low metastatic potential and ICP levels, from patients at high risk, with low metastatic potential ICPs [50], once again highlighting the importance of M2 TAM-associated proteins in establishing an immune score with predictive or prognostic meaning. Starting from these data and applying ssGSEA to a training set and a validation set (GSE21257 dataset), a stratification into an H-group (with high immune infiltration, good response to immunotherapy, high human leukocyte gene expression, good prognosis) and an L-group (with low immune infiltration, poor response to immunotherapy, low human leukocyte gene expression, poor prognosis) was built and validated [51], enforcing the three-gene signature previously indicated.

In partial contrast with the previous studies, however, using the same tools on a different database of HGOS patients, other authors demonstrated that M1 macrophages were associated with poor PFS (progression-free survival) and with the high expression of ICPs such as PD-L1, PDCD1LG2 (Programmed Cell Death 1 Ligand 2), CTLA4 (Cytotoxic T-Lymphocyte Associated Protein 4) and TIGIT (T-cell immunoreceptor with Ig and ITIM domains) in HGOS TIME. This study also identified other TAM-associated biomarkers—namely IL10, VAV1 (Vav Guanine Nucleotide Exchange Factor 1), CD14 and CCL2—as predictors of high-risk versus low-risk patients [52].

Besides the studies that focused on TAM-associated genes, one study demonstrated that an IRG (inflammation regulatory gene) signature, which also involved TILs and other immune-infiltrating populations, constituted an independent prognostic factor and allowed for the identification of specific deregulated processes in immune infiltration and innate immunity [53] that are responsible for the different patients’ outcomes. This result was indirectly confirmed in an independent study based on a retrospective cohort of 79 HGOS patients and a validation cohort of 82 HGOS patients, reporting that innate immunity genes were differentially expressed in low-risk patients [54].

Meanwhile, most studies aiming at building an immune score were focused on inflammation- or immune activity-related genes, and a limited number of studies were focused on oncogenes or oncosuppressor genes present in HGOSs, as well as on serum biomarkers, to build an immune score. In one of these studies, for instance, the Oncomine database identified the trio of p16, p53 and PCNA (proliferating cell nuclear antigen) as being differentially associated with high or low infiltration of TILs, and consequently, different OS (overall survival) [55]. This is the first study showing that genetic variations in tumor genes—namely p16, p53 and PCNA—have not only an impact on HGOS cell proliferation/apoptosis, but may also affect HGS TIME composition. The mechanism underlying this event is not described in this work: we may speculate that HGOSs with mutated p16, p53 and PCNA secrete soluble factors that can prevent the infiltration of specific populations, as TILs.

Recently, ncRNAs (noncoding RNAs) have emerged as nonprotein biomarkers that may have a role in the onset, progression and drug resistance to treatment. circRNAs (circular RNAs), which are a ring protein-non-coding RNA highly conserved in evolution, and miRNAs (microRNAs), which are 20–22-nucleotide-long RNAs derived from introns or protein-coding genes and are involved in protein expression at a transcriptional and post-transcriptional level, are among the ncRNAs most studied as predictors of tumor evolution or as a response to treatments in cancer [56,57]. Only one study has been carried out in HGOS, obtaining interesting results. Microarray data obtained by the GEO datasets identified more than 150 circRNAs and miRNAs, as well as mRNAs (messenger RNAs), in the patients’ serum, correlating with the immunophenotype and the metastatic risk of HGOS [58]. This work indicated that not only immune-associated proteins, but also RNAs involved in their gene regulation, may have a clinical significance, adding another piece to the puzzle to build an immune score for HGOS.

Based on the reported studies, several contradictory findings emerge. One reason explaining these discrepancies could be the use of different patients’ databases, and the fact that those for HGOS are usually small compared to other tumor types [59], given the rarity of the disease. To achieve a significant sample size, patients with different ages, grades, pathological and molecular phenotypes are often included in different databases, or even in the same database used for a specific study. This introduces a bias in the comparability of the patients’ features, diminishing the reliability of the experimental findings. A second reason could rely on the intrinsic intratumor heterogeneity [23], genetic instability and different evolution of HGOSs that unequivocally produce poorly comparable data. Meta-analyses or studies on larger and more homogeneous cohorts of patients may lead to the reliable identification of robust prognostic immune biomarkers. Therefore, the immune score identified until now should be considered provisional; only once the biases emerged have been solved we can reasonably draw an immune signature for HGOS that is predictive of tumor progression or response to treatment, as has already been designed for other more frequent and homogeneous tumors.

## 3. Mechanisms of Resistance to the Host Immune System

After the overview of the immune cell populations infiltrating the HGOS TIME, in this section we will discuss the factors that further impair the activity of antitumor populations and boost the functions of the immunosuppressive cells. We will mainly focus on two parameters characterizing HGOS TIME: the hypoxic environment and the mechanisms hampering the chemotherapy-induced ICD of tumor cells.

### 3.1. Hypoxia Induces an Immunosuppressive Environment in Osteosarcoma

A factor that may affect the activation of the host immune system against HGOS cells is hypoxia, which is well documented in HGOSs by the positivity of hypoxic-related biomarkers such as HIF-1α and target genes such as GLUT-1 (glucose transporter 1), CA IX (carbonic anhydrase IX) or VEGF, whose high levels are associated with lower overall survival and PFS in patients [19,60]. Being physiologically present during normal bone morphogenesis, where it promotes osteogenesis and bone resorption by osteoclasts, in HGOSs, hypoxia favors EMT (epithelial mesenchymal transition), promoting higher invasiveness and stemness maintenance and endorsing tumor recurrence and drug resistance [61]. Hypoxia generates an immune desert in solid tumors, by d ecreasing the ratio between antitumor M1 TAMs and tumor-permissive M2 TAMs. This event in turn inhibits the phagocytic properties of DCs and the tumoricidal properties of NK cells and CD8^+^T lymphocytes, by upregulating PD-1. In addition, hypoxic tumors release lactate and chemokines such as CXCL12, CCL5 (chemokine (C-C motif) ligand 5) and CCL28 (chemokine (C-C motif) ligand 28), which recruit immunosuppressive populations such as Treg cells and MDSCs [62]. Moreover, the avid consumption of glucose by hypoxic tumors contributes to immunosuppression because it subtracts the critical fuel of proliferating T lymphocytes, inducing their anergy [62]. In HGOSs, it is well documented that the hypoxic environment impairs the polarization of TAMs, favoring M2 TAMs as prevailing macrophages that are activated by the immunosuppressive IL-10 and IL-4 cytokines, and by the PI3K/Akt/mammalian target of rapamycin (mTOR) axis in HGOS cells [61] that release unidentified soluble factors promoting M2-polarization. M2 TAMs inhibit the tumoricidal activity of CTLs by further releasing IL-6 and IL-10, producing TGF-β, which favors EMT, releasing VEGF and PDGF (platelet-derived growth factor), which induce angiogenesis and HGOS cell invasion [62]. In addition, hypoxia dramatically reduces the number of infiltrating TAMs [63], a barrier against the metastatic HGOS cells [20], and also favors the production of VEGF and MMP-9 by MDSCs [9], further promoting tumor growth and invasion.

### 3.2. Resistance to Immunogenic Cell Death Induced by Chemotherapy in Osteosarcoma

ICD is a type of cell death that promotes the phagocytosis of dying tumor cells by DCs, followed by the activation of antitumor CD8^+^ T lymphocytes [64,65]. Capsaicin, an inflammatory product of chili pepper [66], and doxorubicin, one of the first-line drugs in HGOS treatment, are strong inducers of ICD [64]. However, the efficacy of chemotherapy-induced ICD is determined by different factors. For instance, cells resistant to doxorubicin, including HGOS, are also resistant to ICD, for multiple reasons.

First, ABCB1 lowers the intracellular accumulation of doxorubicin to a concentration insufficient to determine ICD [67]. Second, ABCB1 inhibits the immune-sensitizing functions of calreticulin, an ER (endoplasmic reticulum)-residing protein that migrates to the plasma membrane after chemotherapy or ER stress, and triggers DC-mediated phagocytosis [68]. Third, the more HGOS is resistant to chemotherapy, the less it suffers from ER stress, a condition determining a huge burden of unfolded proteins within ER lumen, which is necessary to trigger ICD [69]. Since these features represent resistance factors, they may be also exploited as vulnerabilities. For instance, we designed and validated a mitochondria-targeted doxorubicin that bypasses the efflux mediated by ABCB1 and triggers a mitochondrial/ER-coupled stress that results in ICD [67]. A second pharmacological tool has been represented by a synthetic doxorubicin that accumulates within the ER, where it induces massive unfolding, followed by ubiquitination of different proteins, among which is ABCB1 [69]. For this reason, such synthetic doxorubicin also obtained a good antitumor efficacy in preclinical models of HGOSs rich in ABCB1, reinducing ICD [70].

By analyzing the TARGET (Therapeutically Applicable Research to Generate Effective Treatments) and GEO databases, chemotherapy responder or nonresponder patients were sorted out and evaluated by ssGSEA and CYBERSORT algorithms for immune-related genes. Among more than 200 DEGs, 5 genes—TNFRSF9 (TNF Receptor Superfamily Member 9), CD70, EGFR, PDGFD (platelet-derived growth factor D) and S100A6 (S100 Calcium Binding Protein A6)—emerged as determinants of poor response to chemotherapy in HGOSs [71]. The high expression of these genes and downstream-related pathways likely explains the lower efficacy of chemotherapy through a poor induction of ICD, and might be considered a signature predictive of ICD resistance. Using a similar approach, a set of genes related to unfolded protein response (UPR) that trigger ER stress—STC2 (Stanniocalcin 2), PREB (Prolactin Regulatory Element Binding), TSPYL2 (testis-specific protein Y like 2) and ATP6V0D1 (ATPase H+ Transporting V0 Subunit D1) [72], together with ATG16L1 (autophagy related 16 like 1 gene) [73], were identified as genes significantly associated with a high infiltration of antitumor CD8^+^ T lymphocytes in HGOSs. Since UPR and autophagy were necessary to trigger ICD [65], we may hypothesize that ATG16L1 is a factor predictive of good ICD.

Recently, we described a novel mechanism of evasion from chemotherapy and ICD: ABCB1-expressing HGOSs had low expression of ABCA1 (ATP Binding Cassette Subfamily A Member 1) transporter, which effluxes IPP (isopentenyl pyrophosphate), an upstream cholesterol metabolite that activates the tumor-killing properties by Vγ9Vδ2 T cells [74]. Therefore, HGOSs with a high ABCB1 and low ABCA1 phenotype result, at the same time, in chemoresistance and being prone to the evasion of immune killing [75]. However, this phenotype is druggable, because the Ras/ERK1/2/HIF-α transcriptional axis induces ABCB1, while the Ras/Akt/mTOR axis transcriptionally represses ABCA1. Zoledronate, an aminobisphosphonate that inhibits farnesyl pyrophosphate synthase and downregulates Ras-dependent pathways, tips over the balance between ABCB1 and ABCA1, restoring the sensitivity to doxorubicin and immune killing by Vγ9Vδ2 T cells [75]. This work provides an example of the possibility to simultaneously reverse chemo- and immune resistance in HGOSs.

Overall, these few examples demonstrate that the molecular pathways that determine immune resistance (e.g., resistance to ICD) or suppression of tumoricidal functions of CTLs can also be exploited as an Achilles’ heel of HGOS immune resistance by using synthetic chemotherapeutic drugs and/or targeted therapies that undermine the circuitries, inducing immune suppression and restoring the sensitivity to the host immune-system-mediated tumor cell killing.

## 4. Mechanisms of Resistance to Immunotherapy

The TIME of HGOS, where immunosuppressive populations prevail, and the specific features of the HGOS niche, such as hypoxia and resistance to ICD (described in Section 2 and Section 3), explain why HGOSs are often refractory to the immune recognition/immune killing of the host immune system. At the same time, the same features may explain the resistance that adoptive immunotherapy such as ICIs and CAR T cells encountered against HGOSs, as is critically reviewed below. On the other hand, a good understanding of the bases of resistance to immunotherapy may open new experimental ways to enhance immunotherapy efficacy.

### 4.1. Resistance to Immune Checkpoint Inhibitors

The treatment with ICIs induces a relatively low increase in the immune infiltrate, in the T-cell clonalities and in the neoantigen appearance [76], indicating an intrinsic resistance of HGOSs to this type of immunotherapy. The induction of lymphocytic anergy is typically associated with aggressive forms of HGOS. Indeed, in both patients and immune xenografts, it has been shown that the increase in PD-1 and TIM-3 on CD8^+^ CTLs are typical of HGOS progression [77].

Advanced HGOSs are more prone to induce an immunosuppressive environment than early-stage HGOSs: indeed, HGOS cells metastasizing in the lungs have more abundant PD-L1, TIM-3 and LAG-3, and TILs infiltrating metastatic lesions have higher amounts of PD-1 and LAG-3, and lower production of IFN-γ than TILs in primary sites [78]. Consistently, the presence of PD-L1 and LAG-3 is associated with bad PFS [78]. High levels of PD-L1 are strongly correlated with a metastatic phenotype [79], indicating that HGOS inevitably progresses when the host immune system is anergic.

Several mechanisms at the basis of the upregulation of ICPs or their ligands in HGOSs have been proposed. For instance, metastatic HGOS cells secrete high levels of soluble FZD2 (Frizzled-related protein 2), which activates noncanonical Wnt pathways, producing an increase in intracellular Ca2+. The latter, in turn, activates NFAT (nuclear factor of activated T cells) in TILs [80]. Among the targets of NFAT, there is the ectoenzyme CD38, whose down-stream signaling reduces the expression of PD-1, reducing the efficacy of anti-PD-1/anti-PD-L1 treatment [81]. Furthermore, exosomes released by HGOS cells contain PD-L1, and can mediate a horizontal transfer of this ICP ligand, accelerating the tumor progression [82]. Interestingly, PD-L1 level is not only correlated with metastases but also with resistance to cisplatin [83], drawing a parallel with chemoresistant HGOS cells, which are also resistant to ICD. The association with chemoresistance seems to be caused by the low levels of the antimetastatic miR-519d-3p, which destabilizes PD-L1 mRNA. The overexpression of this miRNA reduces PD-L1 levels, lung metastases and resistance to cisplatin at the same time [83], offering novel opportunities for a miRNA-based multitarget treatment of HGOS.

Moreover, the crosstalk with other immunosuppressive populations can increase ICPs in HGOSs. For instance, the high number of MDSCs that limit the expansion and activity of CD8^+^ CTLs also reduce the amount of PD-L1 on HGOS cells. Although no soluble mediators have been identified, it has been demonstrated that inhibiting PI3Kδ/γ (phosphatidyl inositol-3 kinase δ/γ) in MDSCs increases the level of PD-L1, favoring the synergistic antitumor effect of ICIs [84]. Consistently, the anti-CD40 antibody, which re-induces costimulatory receptors on myeloid-derived cells infiltrating the tumors, disrupts the bad liaison between MDSCs and T lymphocytes, reducing the number of PD-1^+^ TILs and the amount of Treg cells, and synergizing with the anti-PD-1 antibody [85]. PD-L1 present on HGOS also blunts the tumoricidal activity of NK cells, by preventing the secretion of granzyme B [32]. These data suggest that this ICP ligand simultaneously inhibits multiple antitumor immune cells.

As has already been carried out for IRG signatures, ICP-based specific immune scores were also proposed. One recent study correlated the expression of RBPs (RNA-binding proteins) with the response to ICIs, since RBPs are known to promote cell proliferation, metastases and drug resistance in several tumors. Exploiting the TARGET, GTEx (Genotype-Tissue Expression) and GEO databases, followed by the ssGSEAS and ESTIMATE algorithms, it was possible to associate the low expression of ICPs with a specific RBP-predicting signature, indicative of worse overall survival [86]. These results can be explained by the ability of RBPs to control the levels of ICPs at translational/post-translational level; the reduced expression of ICPs, although beneficial in relieving TILs’ anergy, also reduced the efficacy of ICI-based immunotherapy, justifying the worse overall survival of the treated patients.

Finally, epigenetic events may also regulate the response to ICIs: indeed, PD-1 is upregulated by the chromatin-remodeling bromodomain-containing proteins [87]. Consistently, the high-throughput analysis of different methylation sites in 35 patients with HGOS indicated that patients can be divided into two methylation clusters associated with good or poor response to anti-PD-1 agents, respectively [88].

### 4.2. Resistance to CAR T Cells

CAR T cells have been used as the most promising cancer immunotherapy, having an antitumor efficacy primarily demonstrated in hematological diseases. More recently, the use of CAR T cells has also been explored in the treatment of HGOS [89].

One of the major problems in the treatment of CAR T cells in solid tumors is to find an ideal targetable antigen. For most solid tumors, it is common to find a tumor-associated antigen that is enriched only on tumor cells and present at low levels on normal tissues. Based on the expression of the target antigens, CAR T cells directed against Her2, GD2 (disialoganglioside 2), B7-H3 and NKG2D have been the most tested engineered T cells against HGOSs [9].

HGOSs are considered Her2 low-expressing tumors; therefore, they are not treated with anti-Her2 monoclonal antibodies. However, the constitutive levels of Her2 represent a good starting point to develop Her2 CAR T cells [90,91].

GD2 is abundant in HGOS cells and is maintained during tumor progression [91], but CAR T cells against this antigen fail in controlling HGOS xenografts, except when combined with all-trans-retinoic acid, an agent that relieves the immunosuppression elicited by MDSCs [92]. This negative result suggests that GD2 alone is not a sufficiently immunogenic target to antagonize HGOS growth. Interestingly, bispecific CAR T cells against Her2 and GD2 obtained a good control of HGOS growth in murine models. The antitumor activity of these bispecific CAR T cells was enhanced in the presence of anti-PD-1/PD-L1 antibodies [93], confirming the hypothesis that CAR T cells against GD2 must be accompanied by additional immunotherapy treatments to achieve a good efficacy. Moreover, the cytoarchitecture of HGOS may also reduce the efficacy of GD2 CAR T cells; indeed, it has been shown that the expression of GD2 in vitro depends on the confluence of cells [94]. Although it is difficult to draw a parallel between the in vitro and in vivo situation, we cannot exclude that differences in HGOS cellularity or architecture impact on this class of CAR T cells.

B7-H3/CD276 is another antigen common to several solid cancers, expressed in primary and metastatic HGOS. Notably, B7-H3/CD276 is highly expressed on the surface of HGOS cell lines, but it is not present on normal peripheral cells or tissues. Indeed, CAR T cells engineered against this antigen successfully reduced the growth of an HGOS orthotopic xenograft, as well as metastatic dissemination [95].

Both primary and metastatic HGOS express NKG2D ligands. This observation, coupled with the finding that activated NK cells that express NKG2D receptor are excellent cytolytic cells against HGOS cells in vitro, led to the development of NKG2D CAR T memory cells that showed good antitumor activity against murine HGOS models [96].

Furthermore, other CAR T cells, such as those engineered against IGF1R (Insulin-Like Growth Factor 1 Receptor) and ROR1 (Receptor Tyrosine Kinase Like Orphan Receptor 1) receptors, which determine cell proliferation and migration and are both expressed in HGOS, have been produced. In HGOS xenografts, with tumors either localized or disseminated, these CAR T cells showed a good antitumor activity, as indicated by the reduction in tumor volume and the release of IFN-γ [97]. Finally, CAR T cells also directed against ALCAM/CD166 (activated leukocyte cell adhesion molecule), expressed in up to 90% of HGOSs, demonstrating an effective cytolytic activity in vitro and in vivo against HGOS cell lines [98].

Notwithstanding the promising results obtained at the preclinical level, the therapy based on CAR T cells also encounters resistance issues. First, CAR T cells have a limited persistence in the body, and therefore a limited amount of time to reach and destroy malignant cells [99]. The microenvironment of HGOS, characterized by a strongly compact and poorly vascularized matrix, constitutes a barrier for CAR T cells’ penetration [100], further impairing their ability to reach the target cells. Second, the abundance of several tumor-tolerant or frankly immunosuppressive populations, such as M2 TAMs, Treg cells and MDSCs in the HGOS environment, makes the antitumor work of CTLs such as CAR T cells harder [100]. Third, the hypoxic environment of HGOS limits the glucose use by CAR T cells, which is necessary for their expansion, and instead creates an abundance of lactate, which suppresses the effector functions of T lymphocytes [62].

In the attempt to overcome the uncertain efficacy of CAR T cells due to these mechanisms of resistance, the specific limitations in their production (i.e., a long period required for their expansion) and the severe side effects, a recent work proposed to directly engineer PBMC (peripheral blood mononuclear cells) against the cell-membrane-anchored and tumor-targeted attIL12; this attempt was effective in reducing the growth of HGOS xenografts, activating a tumor immune killing and a differentiation of HGOS cells [101], representing a new possibility to be explored in the future at a clinical level.

### 4.3. How to Restore Immune System Activity and Immunotherapy Efficacy against Osteosarcoma: Biological Bases

Based on HGOS TIME [9], several strategies attempting to reverse tumor-induced immunosuppression have been tested preclinically. Since one of the most abundant populations infiltrating HGOS and determining tumor progression is constituted by M2 TAMs, the first studies have been focused on them. For instance, all-trans-retinoic acid has been used to reduce the production of IL-4 and IL-13 by M2 TAMs: the reduction in these two cytokines decreased the percentage of cancer stem cells, reducing HGOS growth both in vitro and in vivo [15]. A second promising target is CSF1R (colony stimulating factor 1 receptor), which is abundant in TAMs: its pharmacological inhibition using Pexidartinib reprogrammed the TAM polarization and increased the infiltration and activity of the T lymphocytes, reducing the growth of primary HGOSs and lung metastasis in animal models [102].

A second immune population that has been considered a good target are MDSCs. CXCR4 (C-X-C Motif Chemokine Receptor 4) is a typical receptor expressed by MDSCs: the CXCR4 antagonist AMD3100 effectively decreased the number of MDSCs and was synergic with anti-PD-1 antibody [103], indicating an effective relief of the immunosuppression of MDSCs on T cells. A similar effect was achieved by the pharmacological inhibition of PI3Kδ/γ, abundant in TAMs and MDSCs, combined with the anti-PD-1 antibody [84]. Unluckily, MDSCs do not have unique targets; therefore, the inhibitors proposed are often unspecific, raising concerns about their safety in patients.

Using a complementary approach, several works have increased killing activity of tumoricidal populations, by activating the phagocytic capacity of DCs with stimulating agents such as capsaicin [66] or anti-TGF-β antibodies [104], and/or using DC-based vaccines, based on tumor antigen-loaded CD103^+^ DCs or on CD1^+^ DCs combined with the immune-adjuvant poly I:C (polyinosinic:polycytidylic acid) [105]. Acting downstream of DCs, the activation of CD8^+^T cells has been achieved by TLR4 agonists [30] or CSF1R antagonists [102] that have proved to promote CD8^+^ T-cell activation in OS murine models. The combined use of different ICPs, alone or combined with radiotherapy [106], or of ICPs plus L-arginine, an essential aminoacid for T lymphocytes, increased CD8^+^ T-lymphocyte infiltration and tumoricidal activity, as demonstrated by their increased release of IFN-γ, granzyme B and perforin [107].

Activation of CD8^+^ T cells is certainly an effective strategy, but only if it is preceded by an effective ICD of HGOS cells. The use of mitoxantrone combined with the HSV-1 ICP0 null oncolytic virus KM100 [108], as well as the use synthetic doxorubicins [67,69], are examples of effective strategies of killing HGOS cells by enhancing ICD and CD8^+^ T-cell activity. Interestingly, noncompetitive ABCB1 inhibitors promoting the protein’s ubiquitination act as potent ICD inducers in solid tumors [109]. Although they have not been tested in HGOS yet, they may represent dual-target drugs that are able to increase the efficacy of chemotherapeutic substrates of ABCB1 and to trigger ICD. Moving to a larger scale, a high-throughput screening of the NCI-repository Mechanistic Diversity Set has led to the identification of cardiac glycosides and septacidin, an antibiotic derived from *Streptomyces fibriatus*, as inducers of ICD in different cell models, including human HGOS [110]. Moreover, physical methods, such as hyperthermic treatment by microwaves [111], induce ICD, and could be considered in a translational perspective in the future.

In recent years, technological advances in nanomedicine have shown a good potential to reactivate ICD against refractory tumors. Among the nanomedicine-based tools, manganese dioxide nanoparticles—coated with K7M2 membrane to increase the active targeting and carrying ginsenoside Rh2 and alendronate, which, similarly to zoledronate [112], induce ICD—increased the antitumor activity of CD4^+^/CD8^+^ TILs [113]. Moreover, these nanoparticles decreased immunosuppressive Treg cells and increased the lymphocytic synthesis of proinflammatory cytokines IL-6, IFN-γ and TNF-α [113], thus acting on multiple features of HGOS immune evasion. Similarly, synthetic curcumin-loaded nanoparticles, which stimulate autophagy in HGOS cells, have revealed immunogenic properties, by increasing DC infiltration and maturation as well as CD8^+^ T-lymphocyte activation [114]. Interestingly, these nanoparticles had a synergistic effect with anti-PD-1/PD-L1 [114], opening the possibility of combinatorial therapy. Similarly, the use of mitochondria-targeting micelles carrying the pyruvate dehydrogenase kinase inhibitor 1 dichloroacetate induced a pyroptotic death of HGOS cells, which is highly immunogenic. Notably, pyroptosis releases PD-L1 in a soluble form, relieving the T-lymphocyte anergy [115]. In addition, this strategy could be useful to sum the benefit of restoring the efficacy of ICD and ICIs.

Recently, it has been pointed out that oncolytic viruses can increase the efficacy of chemo-immunotherapy, including ICIs. This was also proven in HGOSs, where the telomerase-specific replication-competent oncolytic adenovirus OBP-502 was used against murine HGOS xenografts, with a low rate of ICD and rich in PD-L1. The association with an anti-PD-1 increased the efficacy of the latter and the infiltration of CD8^+^ TILs, likely because this combination produced a massive autophagy, restored ICD and upregulated PD-L1, favoring the activity of anti-PD-1/anti-PD-L1 agents [116].

If several studies demonstrated the possibility to reactivate ICD in HGOS and obtain a good synergism with ICIs, only indirect observations suggest that this type of approach can work synergizing with CAR T cells. For instance, graphene oxide nanoparticles conjugated with the anti-Her2 Trastuzumab generated a strong reduction in HGOS growth caused by oxidative stress and necroapoptosis, a type of highly immunogenic cell death [117]. The use of these nanoparticles could be combined with anti-Her2 CAR T cells, opening the way to innovative immunotherapy-based combinatorial approaches.

Overall, multiple circuitries in HGOS TIME can be targeted to reverse the immunosuppression induced by the tumor and improve the efficacy of either the immune killing of the endogenous host immune system or the efficacy of immunotherapy. This knowledge has been translated in specific clinical trials, which are critically discussed in the last section of this review.

## 5. Immunotherapeutic Treatment Strategies

Treatment strategies based on immunotherapeutic approaches can be subgrouped into different categories (Figure 3), which are briefly described in the next sections.

### 5.1. Tumor-Associated Macrophages (TAMs)

Macrophages tend to infiltrate and accumulate in tumor tissues and TIME, and the so-called TAMs have been indicated to participate in tumor progression [118,119,120]. Although TAMs represent at least 50% of immune cells that are present in HGOS TIME [121,122], their role and involvement in the progression of this tumor remain controversial and are still under discussion [2]. However, due to their relevant presence in HGOS TIME, clinical trials involving TAMs have been performed or are presently ongoing (Appendix A). These trials involved the use of L-MTP-PE (liposomal muramyl tripeptide phosphatidylethanolamine; mifamurtide), a synthetic drug that stimulates immune responses and activates macrophages and monocytes [3,123]. L-MTP-PE has been suggested to influence HGOS progression by affecting M1/M2 polarization of TAMs [124,125,126] and has provided evidence of a manageable safety profile [127].

Mifamurtide was first tested in phase I and II clinical trials, enrolling patients with metastatic or relapsed HGOS [126,128,129]. These trials, in which patients were treated with or without L-MTP-PE in combination with multidrug adjuvant chemotherapy, achieved a significantly higher overall and disease-free survival in the subgroup of HGOS patients receiving L-MTP-PE. Moreover, these regimens demonstrated that L-MTP-PE and chemotherapy can be effectively used together, and encouraged the planning of additional clinical studies.

The phase II/III trial (NCT01459484) launched in 2011 by the Italian Sarcoma Group (ISG) provided evidence for a benefit in using mifamurtide in association with chemotherapy in selected high-risk groups of HGOS patients [130]. In this protocol, based on the evidence that in HGOSs, overexpression of ABCB1 at diagnosis was associated with a worse outcome [131,132,133], mifamurtide was used postoperatively to intensify chemotherapy for patients with ABCB1 overexpression at clinical onset. The trial recruited a total of 279 patients and closed in March 2018. Results showed that adjuvant mifamurtide, combined with high-dose ifosfamide, can improve event-free survival in HGOS patients with ABCB1 overexpression at diagnosis and poor response to preoperative chemotherapy, suggesting that this regimen might be proposed as salvage therapy for HGOS patients unresponsive to standard treatments [130].

The phase II EuroSarc-Memos trial (NCT02441309) enrolled patients with recurrent and/or metastatic HGOS, who were treated with mifamurtide, either alone or combined with ifosfamide [134]. Unfortunately, the trial was early terminated due to a poor recruitment within the allocated funding period, without providing conclusive results.

TAMs can also be activated by targeting CD47, which inhibits phagocytosis of nontumor cells by binding SIRP and impairing its signaling pathway in macrophages. Since tumor cells can use the same mechanism as a tool for immune evasion, anti-CD47 monoclonal antibodies have been proposed as anticancer agents aimed to reactivate TAMs in the TIME [135]. Based on the evidence that CD47 is highly expressed in HGOS and inhibits the tumor progression in HGOS xenograft models [136], clinical trials in which both CD47 and GD2 are targeted with monoclonal antibodies are under evaluation.

### 5.2. Immune Checkpoint Inhibitors (ICIs)

Therapies based on ICIs targeting the PD-1/PD-L1 pathway have provided promising clinical results in adult patients with advanced cancers and have also been considered for HGOS (Appendix A).

More recently, in order to overcome the limitations of single-ICI regimens, new treatment schedules have combined therapies directed against PD-1, PD-L1 and the CTLA-4 or with a second nonimmunological target [137,138]. Trials based on this approach are presently ongoing (Appendix A) and may hopefully open new treatment perspectives for HGOS.

Some ICI-based clinical studies in HGOSs have been completed and have provided results.

The NCT02301039 trial reported an objective partial response in 1 of 22 patients and a disease stabilization in 6 out of 22 patients with HGOS who were treated with pembrolizumab, an anti-PD-1 monoclonal antibody [139].

Another single-arm, phase II trial (NCT03013127) based on the use of pembrolizumab enrolled 12 patients with advanced HGOS. This study unfortunately did not show any clinical benefit at 18 weeks of treatment, leading to termination of the trial [140]. Authors concluded that despite pembrolizumab resulting in being well tolerated, the lack of evidence of clinical antitumor activity in advanced HGOS patients suggested that future immunotherapy treatments should explore combination strategies, possibly in patients stratified according to molecular profiles. This trial, however, provided evidence similar to other studies that had been previously conducted, confirming that the use of single-agent ICP inhibitors in advanced HGOS is poor or not effective [141,142]. Based on these indications, and on some evidence showing that blocking PD-1/PD-L1 in combination with a second target seems to be more effective than monotherapies, new combined treatment regimens have also been explored in HGOS.

In the NCT03359018 trial, camrelizumab (a PD-1 inhibitor) was used to treat advanced HGOS patients in combination with apatinib, a tyrosine kinase inhibitor that selectively inhibits the VEGFR2 (vascular endothelial growth factor receptor-2, also known as KDR). The combined treatment seemed to improve progression-free survival compared with the arm using apatinib alone, but additional studies are needed to further support the hypothesis that this might be considered as a new promising therapeutic perspective for HGOS [143].

The NCT02815995 trial evaluated the efficacy, safety and changes induced in TIME by durvalumab (an anti-PD-L1 drug) in combination with tremelimumab (an anti-CTLA-4 drug) in advanced or metastatic soft tissue and bone sarcomas [144]. This study revealed a histology-specific response, and unfortunately, all of the five enrolled HGOS patients experienced an adverse outcome. However, based on these results, the authors concluded that the combination of durvalumab and tremelimumab is active in advanced or metastatic sarcoma patients, and warrants further evaluation in future trials for specific sarcoma types.

### 5.3. Targeting Disialoganglioside 2 (GD2)

GD2 is a surface protein that is widely expressed in HGOS [145,146], which has been explored as a possible therapeutic target in a few clinical trials [147,148], some of which are still ongoing (Appendix A). The advantage of targeting GD2 resides in the fact that its expression is limited to very few normal tissues (mostly cerebellum and peripheral nerves), indicating that this membrane molecule may represent a rather specific tumor therapeutic target.

Targeting GD2 has been also combined with other treatments. For example, in the ongoing phase II trial (NCT02502786; Appendix A), the humanized monoclonal anti-GD2 antibody 3F8 (hu3F8) is combined with granulocyte-macrophage colony stimulating factor (GM-CSF), in order to simultaneously stimulate the activity of TAMs with the aim of improving the efficacy of GD2 targeting alone.

Recently, preclinical findings showing that dinutuximab, a monoclonal antibody targeting GD2, can exert relevant antitumor effects on HGOS cell lines and xenografts [93] provided the rational for the design of additional GD2-based clinical trials (Appendix A). Unfortunately, the AOST1421 (NCT02484443) phase II trial, in which dinutuximab was used in association with GM-CSF, did not show evidence of outcome improvement in a series of 39 patients with recurrent pulmonary HGOS in complete surgical remission [149], demanding further exploration of this treatment approach.

### 5.4. Immunotherapy Based on Immunomodulating Agents

Immunomodulation is highly demanded to exploit the immune system of tumor patients at its best, especially toward targets expressed in tumor or immune cells hampering immune response and chemotherapy efficacy against cancer cells. Thus, many endeavors are directed in currently active clinical trials toward such targets [148]. An example is represented by SEMA4D (semaphorin 4D), which is revealed to be a proto-oncogene causing activation of PI3K-AKT-mTOR and MAPK (mitogen-activated protein kinase) signaling. Moreover, SEMA4D was found to possibly enhance tumorigenic properties of human HGOS cell lines when overexpressed, since in physiological conditions SEMA4D is expressed only in osteoclasts, whereas it is absent in osteoblasts [150]. The monoclonal antibody pepinemab directed against SEMA4D is involved in a phase I/II clinical trial in patients with solid tumors comprising HGOS (NCT03320330), although no results have been published to date.

The PI3K-AKT-mTOR pathway itself is a target for therapy because of its pivotal role in controlling cell growth, metabolism and survival in many types of cancers, including HGOSs [151]. For example, mTOR inhibitor sirolimus has been used as an immunosuppressive drug, and many trials include this agent in combination with other drugs or biological therapies in HGOSs [89]. In the phase I/II trial NCT02584647, sirolimus is coupled with PLX3397 (pexidartinib), a CSF-1R monoclonal antibody, to target TAM polarization in different sarcomas. Despite this trial not enrolling HGOS patients, it highlighted the safety of this drug combination and revealed a clinical benefit in 67% of patients together with a decreased proportion of protumoral M2 TAMs in the examined post-treatment specimens [152].

A recent study assessed that indoleamine 2,3-dioxygenase (IDO)-inhibitor indoximod, enhancing presentation of tumor antigens by DCs, and BTK (Bruton’s tyrosine kinase) inhibitor ibrutinib, promoting differentiation of immunogenic DCs, could synergistically suppress BTK-IDO-mTOR axis in DCs, thus enhancing differentiation of inflammatory DCs and promoting T-cell response against cancer, since many human tumors present DCs expressing IDO and BTK [153]. Based on these results, the phase I trial NCT05106296 is currently recruiting patients with solid malignancies for treatment based on a combination of indoximod and ibrutinib with cyclophosphamide and etoposide, with the aim of improving the clinical response provided by chemotherapy coupled with immune stimulation.

### 5.5. Cellular and Adoptive Immunotherapy Not Involving CAR T Cells

The strategy underlying cellular and adoptive immunotherapy is based on the use of cells that help the patient to cause an antitumor response. In this section, we will review the clinical trial not based on CAR T cells, but on host immune cells such as NKs and DCs.

Of the phase II clinical trial NCT02100891, the results of 15 patients, including 1 patient with HGOS, were presented at the ASCO meeting 2020 [154]. Treatment consisted of HLA-haploidentical bone marrow or peripheral blood stem cell transplantation after reduced-intensity chemotherapy and radiation therapy to optimize graft-versus-tumor effects. Haplo-NK cells were purified from nonmobilized donor mononuclear cells and freshly infused seven days after transplantation. Patients were treated with sirolimus to maintain postgrafting immune suppression. Overall, this dual-immunotherapy approach was well tolerated; no patient died from transplant-related causes. The estimated overall survival was better than expected, with a median follow-up of 1.3 years.

The phase II clinical trial NCT03610490 was presented at the ASCO meeting 2019 [155] together with the NCT03449108 trial, but not an accrual update in 2020, as had been foreseen. TILs were manufactured according to a protocol using urelumab, an agonistic anti-CD137 antibody, combined with T-cell receptor activation during TIL expansion. In both trials, patients underwent excisional tumor biopsy for TIL manufacturing followed by a cyclophosphamide and fludarabine lymphodepletion treatment and up to six doses of IL-2 after TIL infusion.

Another strategy using autologous DCs was used in the phase I trial NCT01803152. Results of 19 patients were presented at the ASCO meeting 2021 [156]. Monocytes were collected from patients by pheresis and incubated with GM-CSF plus IL-4 to generate immature DCs, which were then loaded with autologous tumor lysates prepared from the patients’ surgical resection. The DC vaccine was administered intradermally in imiquimod-treated skin to complete DC maturation. Treatment consisted of four weekly DC vaccine injections followed by four monthly “boosters” of tumor lysate. There was no treatment-related dose limiting toxicity, but authors concluded that initial and sustained antitumor activity must be improved.

Other ongoing phase I trials based on different immunoadoptive strategies are listed in (Appendix A) and shortly summarized below.

The phase I trial NCT02508038 was designed to investigate the safety of transplantation with a haploidentical donor peripheral blood stem cell graft depleted of TCR+ and CD19+ cells, followed by five doses of the immunomodulating drug zoledronate.

In the phase I trial NCT04282044, patients undergo leukapheresis to enable the ex vivo generation of autologous CIK (cytokine-induced killer cells). A fixed dose of CIK cells combined with the specified dose of the oncolytic virus CDSR is administered as CRX100 infusion. In this study, the biodistribution of CRX100 will also be evaluated.

In the phase I trial NCT01590069, patients receive aerosolized aldesleukin, which is a recombinant IL-2 that should stimulate or suppress the immune system to stop tumor cell growth.

### 5.6. Immunotherapy Involving CAR T Cells

Finding antigens preferentially expressed in HGOS compared to non-neoplastic cells is of paramount importance to design trials based on CAR T cells. One of these antigens is B7-H3/CD276, and indeed, CAR T cells directed against B7-H3/CD276 are tested in different ongoing trials (Appendix A).

Another trial is focusing on NY-ESO-1, a cancer-testis antigen expressed only in tumor cells. Its expression was found in 31.3% of HGOSs, and the clinical trial (NCT03462316) testing the efficacy of CAR T-cell therapy based on the NY-ESO-1 antigen on bone and soft tissue sarcomas is still ongoing.

Cellular therapy based on CAR T cells can be used either alone or in combination with pharmacological treatment (Appendix A). For instance, in the phase I trial (NCT01953900), the combination of iC9-GD2-CAR-VZV-CTLs with a varicella zoster vaccine and lymphodepleting chemotherapy to monitor the long-term persistence of these cells and their side effects is studied. Another phase I trial (NCT03635632) is based on GD2-CAR T cells added with the gene C7R, which gives a constant supply of cytokine to help them to survive for a longer period. Trials based on this approach are still ongoing and have no results yet.

For patients with recurrent or refractory HGOS, there are several trials ongoing (Appendix A). In the phase I EGFR806 CAR T trial (NCT03618381), T cells from patients have been modified to express an EGFR specific receptor, alone or in addition with the CD19 receptor, that will target and kill EGFR expressing solid tumors. Another trial (NCT05312411) is focusing on the feasibility and safety of the treatment of FITC-E2 (fluorescein-specific) CAR T cells in combination with the folate fluorescein UB-TT170 in recurrent and refractory HGOS. UB-TT170 should attach to tumor cells and flag them in order to attract CAR T cells and kill cancer cells.

## 6. Conclusions

The role of HGOS TIME as a possible tool to define new treatment strategies is an expanding field, which is under investigation both at the preclinical and clinical level. Several approaches trying to reactivate the host immune system or improve the efficacy of immunotherapy are ongoing in preclinical models. The main limitation of these approaches is the simultaneous presence of both protumor and antitumor cell populations within the HGOS TIME, which sometimes share similar receptors, hindering the selective targeting of a single population to relieve intratumor immunosuppression.

A second important factor is the high degree of exhaustion of CTLs and NK cells that impairs their optimal activation without using ICPs, which are effective in preclinical models but produce severe systemic toxicities in patients. In addition, adoptive immunotherapy, based on CAR T cells or γδ T cells, has encountered several limitations caused by the intrinsic nature of the HGOS extracellular matrix and the hypoxic environment.

Finally, most preclinical studies that have achieved good results have been performed on mice, whose immune system is more reactive than the human one, and therefore may provide indications that cannot be directly translated to humans. Humanized mice, i.e., animals bearing a functioning reconstituted human immune system as immune-PDX (patient’s derived xenografts), could partially overcome this issue, although the immune-PDX are costly and time-consuming and do not perfectly reproduce the patients’ immune environment. These issues, however, leave space for preclinical research to set up more specific and flexible immunotherapy approaches based on combinatorial strategies that act on multiple immune targets at the same time.

At a clinical level, several targets and treatment approaches have been explored and are still under evaluation, also taking advantage of the fact that immune-based therapies exert toxicity profiles, which usually do not overlap with those of conventional chemotherapeutic drugs. The different strategies that are presently under investigation, which are generally aimed to abrogate the immune evasion of HGOS cells, will hopefully help to indicate new ways leading to an improvement of the prognosis of this tumor, in particular in cases of a recurrent, refractory advanced disease.

## Figures and Tables

**Figure 1 ijms-24-00799-f001:**
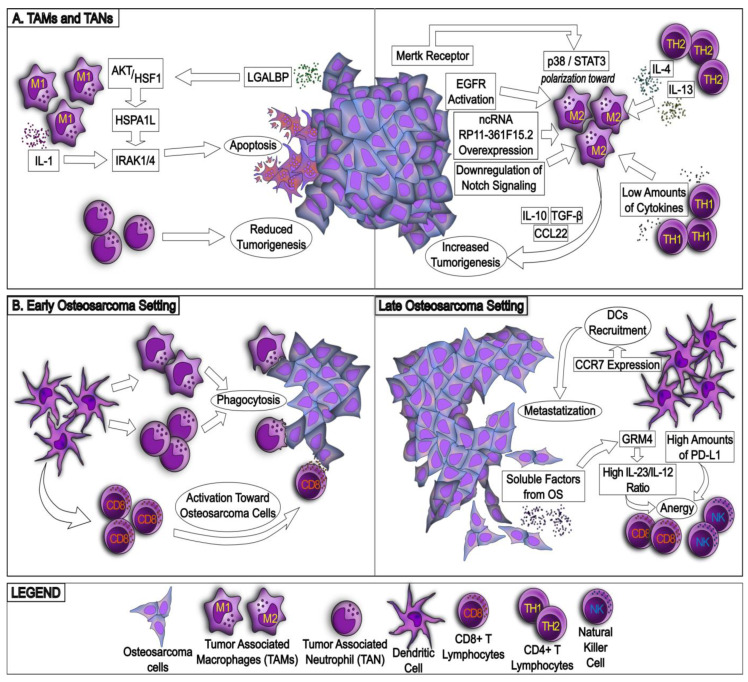
Role of cells of innate immunity present in osteosarcoma tumor microenvironment. (**A**) depicts the role of tumor-associated macrophages (TAMs) and neutrophils (TANs) toward osteosarcoma cells. In particular, whereas TANs have been shown to reduce tumorigenesis, TAMs can be divided in two main categories, which are proved to act by inhibiting (TAM M1) or enhancing (TAM M2) tumor growth of osteosarcoma. In (**B**), the behavior of dendritic cells toward osteosarcoma cells is depicted, which varies depending on tumor setting (early versus late). Molecules and effectors belonging to biological pathways are embedded in squares, and exerted effects are enclosed in circles.

**Figure 2 ijms-24-00799-f002:**
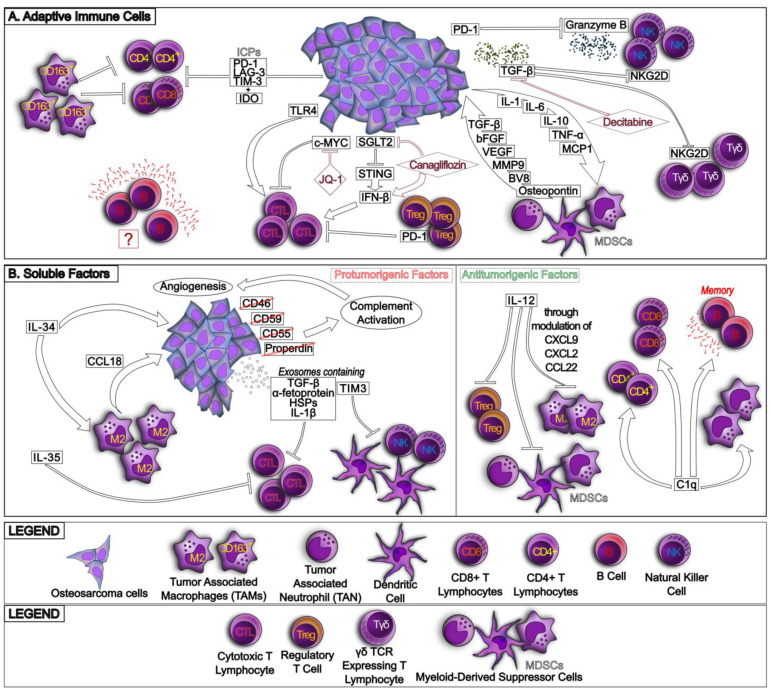
Role of cells belonging to adaptive immunity present in osteosarcoma tumor microenvironment. In (**A**), documented activities of various immune cells toward osteosarcoma are displayed: whereas a role of B cell has not been defined to date, cytotoxic cells and regulatory cells show different behaviors, depending on secreted factors as well as on interplay with innate immunity cells or with myeloid-derived suppressor cells (MDSCs). Documented activity provided by drug inhibitors (brown rhombi) is reported as well. (**B**) depicts different soluble factors and molecules implicated in protumorigenic or antitumorigenic processes. Molecules and effectors belonging to biological pathways are embedded in squares, whereas those with suppressed expression are enclosed in crossed-out squares. Exerted effects are enclosed in circles.

**Figure 3 ijms-24-00799-f003:**
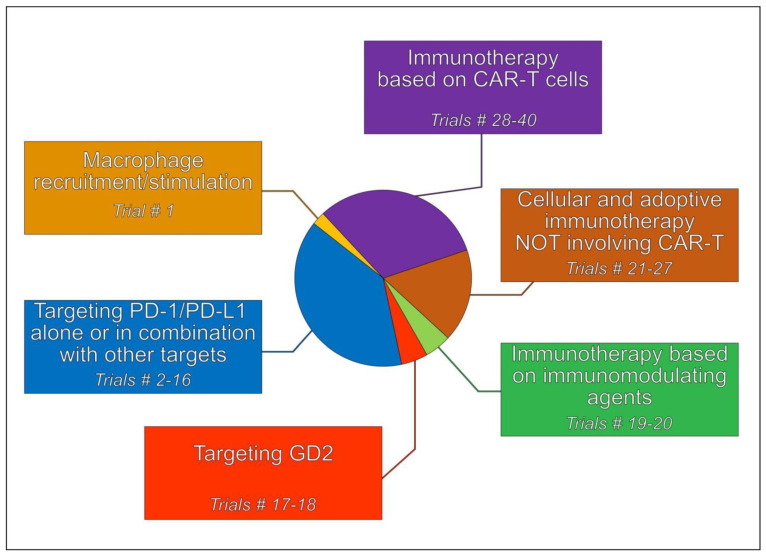
Categorization of clinical trials involving immunological approaches that are currently active in high-grade osteosarcoma (HGOS). Numbers refer to the trials listed in Appendix A.

## Data Availability

Not applicable.

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
