# Peer review of "Strategies to Overcome Resistance to Immune-Based Therapies in Osteosarcoma"

_ijms, 2023, doi:10.3390/ijms24010799_

Round 1

Reviewer 1 Report

This manuscript is a comprehensive review article summarizing the molecular mechanisms behind the osteosarcoma immune environment, as well as challenges and potential solutions to treat high-grade osteosarcoma from both pre-clinical and clinical standpoints. The authors provide wide-ranging references including details in various clinical trials and controversial results for discussion, which is appreciated. In general, the manuscript is well-written and easy to follow. Minor editorial comments are listed below.

1.       Figure 1 legend, line 68: Suggest using “early vs late” but not “early vs advanced” to make consistent with the figure title.

2.       Figure 2B: What is the purpose to cross out CD46, CD59, CD55 and Properdin?

3.       Both Figures 1 and 2 are very busy. Suggest simplifying the contents or removing all the boxes around the molecules.

4.       Lines 80-82: Often time, the reviewer found that the description is not clear regarding the cell source. For example, this sentence described EGF and non-coding RNA, but the reader can be confused by whether these factors were released from tumor cells or immune cells. Such situation should be revised throughout the manuscript.

5.       Line 285: “was” should be “were”

6.       Line 287: Suggest removing “another work”.

7.       Lines 508-511: More information for B7H3 is actually provided in section 5.6. Please make consistent either B7H3 or B7-H3.

8.       Section 4.2 is somewhat overlapping with section 5.6. Might consider reorganizing or consolidating them.

9.       Line 517-522: Suggest rephrasing or breaking into two sentences.

Author Response

Reply to Reviewer 1

Open Review

English language and style

( ) English very difficult to understand/incomprehensible
( ) Extensive editing of English language and style required
( ) Moderate English changes required
(x) English language and style are fine/minor spell check required
( ) I don't feel qualified to judge about the English language and style

Is the work a significant contribution to the field?

Is the work well organized and comprehensively described?

Is the work scientifically sound and not misleading?

Are there appropriate and adequate references to related and previous work?

Is the English used correct and readable?

Comments and Suggestions for Authors

This manuscript is a comprehensive review article summarizing the molecular mechanisms behind the osteosarcoma immune environment, as well as challenges and potential solutions to treat high-grade osteosarcoma from both pre-clinical and clinical standpoints. The authors provide wide-ranging references including details in various clinical trials and controversial results for discussion, which is appreciated. In general, the manuscript is well-written and easy to follow. Minor editorial comments are listed below.

  1. 1.Figure 1 legend, line 68: Suggest using “early vs late” but not “early vs advanced” to make consistent with the figure title.

According to the Reviewer's suggestion, we used “early vs late” instead of “early vs advanced”.

  1. Figure 2B: What is the purpose to cross out CD46, CD59, CD55 and Properdin?

We apologize for not having fully explained why CD46, CD59, CD55 and Properdin were crossed out. The meaning of crossing out CD46, CD59, CD55 and Properdin has been inserted in the revised Figure legend.

  1. Both Figures 1 and 2 are very busy. Suggest simplifying the contents or removing all the boxes around the molecules.

Again we apologize for not having fully described the meaning of the boxes inserted in the Figures. Also this item has been explained in in the revised Figure legends.

  1. 4.Lines 80-82: Often time, the reviewer found that the description is not clear regarding the cell source. For example, this sentence described EGF and non-coding RNA, but the reader can be confused by whether these factors were released from tumor cells or immune cells. Such situation should be revised throughout the manuscript.

We apologize for the lack of clarity. We extensively revised the manuscript; besides the lines 80-82 of the previous version, we specified throughout the manuscript the cellular origin of growth factors or cytokines mentioned.  

  1. Line 285: “was” should be “were

According to the Reviewer's suggestion, we have corrected the text.

  1. 6.Line 287: Suggest removing “another work”.

According to the Reviewer's suggestion, we have corrected the text.

  1. Lines 508-511: More information for B7H3 is actually provided in section 5.6. Please make consistent either B7H3 or B7-H3.

To accomplish the Reviewer request, we have provided more details in section 5.6, when B7-H3 was first mentioned. We have also fixed the acronym as B7-H3 throughout the manuscript.

  1. Section 4.2 is somewhat overlapping with section 5.6. Might consider reorganizing or consolidating them.

We thank the Reviewer for raising this criticism. Indeed, we reorganized paragraphs 4.2 and 5.6.

Paragraph 4.2 now focuses on the biological processes at the basis of resistance to CAR T-cells, whereas paragraph 5.6 describes the clinical trials that are currently based on CAR T-cells.

We also slightly modified the first sentence of paragraph 5.5.

  1. Line 517-522: Suggest rephrasing or breaking into two sentences.

We have changed the text according to the Reviewer's suggestion.

Reviewer 2 Report

The presented review contains comprehensive information about current knowledge of immunosurveillance and immunotherapies used in HGOS.

The review is, unfortunately, very hard to read because of several sections, which are gradually described, but do not connect to each other and the authors often return to them, however, due to the large amount of information, the continuity of the information is lost.

Some part like chapter 2.2. is just a description of results without the putting information together and some kind of discussion or conclusion. Paragraph about RNAs is just an information about their existence without any further information about their role (line 309-314).

It could be more readible to try to keep the information about the particular topic together and put some information on the tables. For examples, re-shaping the chapter 4 about the resistance to introduce the currently used treatments and trials and than explain the resistance to the therapy will provide better continuity to the reader. These trials can be introduced briefly and completed in tables, than the author can be focused on the explanation of resistance. 

The readibility can also be increased by putting the shortcuts first and then the explanation like PD-1 (programed cell death 1). Most of this shortcuts and know and the readers will often skip the explanation (what coul make it easier for them to read).

Line 180 - NK cells are not subset of T cells!

Overal, the review is full of interesting information but in order to show the signs of a constructive analysis of literary data, the structure of the text needs to be redesigned in particular.

Author Response

Reply to Reviewer 2

Open Review

English language and style

( ) English very difficult to understand/incomprehensible
( ) Extensive editing of English language and style required
( ) Moderate English changes required
( ) English language and style are fine/minor spell check required
(x) I don't feel qualified to judge about the English language and style

Is the work a significant contribution to the field?

Is the work well organized and comprehensively described?

Is the work scientifically sound and not misleading?

Are there appropriate and adequate references to related and previous work?

Is the English used correct and readable?

Comments and Suggestions for Authors

The presented review contains comprehensive information about current knowledge of immunosurveillance and immunotherapies used in HGOS.

The review is, unfortunately, very hard to read because of several sections, which are gradually described, but do not connect to each other and the authors often return to them, however, due to the large amount of information, the continuity of the information is lost.

1) Some part like chapter 2.2. is just a description of results without the putting information together and some kind of discussion or conclusion. Paragraph about RNAs is just an information about their existence without any further information about their role (line 309-314).

To make the text easier to be followed, several changes have been made. First, we added a sentence explaining the general scheme of the review at the end of the Introduction. Second, we modified each section to better connect the different issues considered in this review. Third, we added a more critical vision to the sections that we perceived as too descriptive. In particular, we modified the section 2.2. and the paragraph about RNA, according to the Reviewer’s criticisms.

2) It could be more readible to try to keep the information about the particular topic together and put some information on the tables. For examples, re-shaping the chapter 4 about the resistance to introduce the currently used treatments and trials and than explain the resistance to the therapy will provide better continuity to the reader. These trials can be introduced briefly and completed in tables, than the author can be focused on the explanation of resistance.

As suggested by the Reviewer, chapter 4 has partially been rewritten and reorganized.

Information about ongoing trials was already summarized in the Supplementary Table 1 (Table S1) of the first submission, whereas in the text only trials with published results were discussed.

In order to accomplish the Reviewer's indications, in the revised version of the manuscript we have removed the parts of the text that were redundant with Table S1 contents, and we have left only the information which was essential to introduce and explain the rationale of each treatment strategy, highlighting the problem of resistance against each of them.

3) The readibility can also be increased by putting the shortcuts first and then the explanation like PD-1 (programed cell death 1). Most of this shortcuts and know and the readers will often skip the explanation (what coul make it easier for them to read).

According to the Reviewer's suggestion, we have put the shortcuts first and then the explanation of each abbreviation throughout the whole manuscript.

4) Line 180 - NK cells are not subset of T cells!

We apologyze for the mistake that we corrected in the revised version.

5)Overal, the review is full of interesting information but in order to show the signs of a constructive analysis of literary data, the structure of the text needs to be redesigned in particular.

Following the Reviewer's suggestions, we have implemented the discussion of literary data and we extensively reshaped the text to make the connections between the different sections clearer.